# Online Test of Risk Self-Detection of Eating Disorders

**DOI:** 10.3390/ijerph18084103

**Published:** 2021-04-13

**Authors:** Gilda Gómez-Peresmitré, Romana Silvia Platas-Acevedo, Gisela Pineda-García

**Affiliations:** 1Faculty of Psychology, National Autonomous University of Mexico, Av. Universidad 3004 Col Copilco-Universidad, Alcaldía, Coyoacán, Mexico City 04510, Mexico; romsip@unam.mx; 2Faculty of Medicine and Psychology, Autonomous University of Baja California, Cal, Universidad 14418, International Industrial Park, Tijuana 22390, Mexico; gispineda@uabc.edu.mx

**Keywords:** online psychological assessment, risk self-detection, eating disorders, risk predictors, prevention

## Abstract

This study’s central aim was to examine the effectiveness of an online test of the Risk Self-Detection of Eating Disorders through the evaluation of (1) its psychometric properties, the significant probability of predicting risk eating behavior and the correct classification of membership to the risk or non-risk group and (2) the ability to measure users’ risk self-awareness through a group of statements and explore the expected responses through frequency analysis of the content provided by the users. The non-probability sample was comprised of *n* = 541 women aged 15 to 25 (M = 18.3; SD = 2.4). The instrument showed good psychometric properties, a structure of three predictive risk factors, and proper effect sizes (R^2^ = 0.67–0.69). Among the most critical findings were high percentages of correct classification (94–97%) and expected responses (61%). The logistic regression analysis showed that the risk of presenting eating disorders was higher if the participants smoked, consumed alcohol, had sexual experiences before the age of 15, and if those sexual experiences were non-consensual. Evidence is presented about the promising results of the online data collection method and its potential benefits.

## 1. Introduction

There are multiple risk factors involved in the onset and maintenance of eating disorders (ED) in adolescence. Currently, 16% of the world’s population comprises young people (15–24 years old) [1]. In Mexico, a quarter (25.7%) of the total population is adolescent (15–29 years old). Puberty marks the time of the most significant problems concerning dissatisfaction with body image (the girl wants to be thinner, regardless of her actual weight, and the boy wants a robust and muscular figure) [2,3]; in adolescence, this negative attitude is reaffirmed [4]. Adolescence poses one of the most difficult challenges for families, for society, and themselves, since at this stage of life, they face hormonal changes, identity problems, social pressure, sexual behavior, alcohol, drugs, and exposure to violence, among others, all being risk factors that can detonate diseases that affect the ability of adolescents to grow and develop fully [5,6]. Sociodemographic-LifeStyle Variables (SLV) are psychosocial, economic, and sociocultural indicators related to health in general [7] and health-related to the onset and the development of risky eating behaviors (REB) in particular. A high prevalence of REB has been found at early ages, especially in female adolescents (15%), associated with SLV such as smoking, drugs, and alcohol [8,9]. Other SLV are associated with body image (BI) [10], it has also been found that negative BI assessment carries a risk of developing eating disorders, depression, suicidal ideation, risky sexual behavior, and physical violence [11]. Adolescents faced numerous barriers to accessing health products and services that are acceptable, accessible, appropriate, and adequate; hence the importance of encouraging healthy self-care habits in young people [6,12,13] and promoting the use of new health communication technologies. Also, a large sector of the school population (pubescent, adolescent, and young people) do not have access to knowledge, services, and benefits in the area of dietary health and body image [14]. In this regard, [15] point out that despite the significant impact ED has on individuals physically, emotionally, and socio-economically, appropriate services are often not accessed. While adolescents do not have access to health services, the media widely access information about the symptoms of ED (movies, magazines, and others). The adolescents have doubts, they ask themselves, do I have that problem? They have fears (where and with whom to go?) and inhibitions (what will they say, how embarrassing, etc.). Eating disorders have acquired great social and health relevance in recent decades due to their complexity, severity, and difficulty in establishing a diagnosis in all its dimensions. It is a pathology of multifactorial etiology involving biological, family, genetic, and sociocultural factors [16]. Given this severe scenario, there is only one alternative, to prevent risk factors before the emergence of ED by developing efficient solution strategies. Thus, the prevention of risk factors was proposed as a central objective to evaluate the effectiveness of an online test of the Risk Self-Detection of Eating Disorders (RSDED).

An advantage of using online communication/evaluation techniques, specifically the Internet, in the field of psychological research is the confidential and anonymous nature that facilitates communication/evaluation (as opposed to face-to-face), which is aptly called the online disinhibition effect [17]. The anonymity involved in distance evaluation promotes self-awareness and favors results with greater validity [18]. It is essential to point out these characteristics because, as can be seen throughout this paper, disinhibiting communication and promoting self-awareness are precisely the effects that are required for the successful development of the instrument (RSDED) that was evaluated in this study.

Some researchers [19] state that self-help administered by any digital means is an alternative for those who refuse therapy in person, so it can increase the rates of students receiving care for health problems. Self-help/self-detection should be directed according to Eisenberg et al. [20] at removing barriers such as the “unseen need” barrier. Prevention requires self-knowledge of risk, and help is needed, and this is part of what is intended to be achieved with the application of the test (RSDED).

The review of online instruments aimed at adolescents for the purpose of self-detection or self-diagnosis for risk factors to develop ED showed some online programs aimed at a non-community clinical population, looking at symptomatology, not risk factors, offering intervention or treatment, and not just focusing on screening or detecting risk factors in ED. These programs were Appetite for Life [21] based on the Internet providing individual support. European Project Salut [22] and the Student Bodies (SB) [23] are delivered online with therapist moderation. We found other instruments related to ED risk, such as the development of structural models (online) [24] related to academic performance [25] with risk factors in university students [26]. These are all successful programs but are not formatted of self-screening as is RSDED.

The studies conducted to evaluate the effectiveness of distance interventions, e.g., Stice et al. [27], reported that more than a half of risk factor prevention programs for ED had been found to reduce these factors, while almost a third mitigate current or future food pathology. A meta-analysis of online psychological interventions showed results [18] that strongly support the adoption of distance-based psychological assistance activities and are comparable to face-to-face intervention.

The use of the Internet for research/evaluation of psychological problems has been classified into three broad categories: psychological assessment, psychotherapeutic diagnosis, and self-exploration and awareness [18]. Self-exploration/self-awareness (SE/SA) is related to the central objective of this study. The user actively participates, reading/responding to the instrument (RSDED), providing feedback, and exhibiting emotions associated with risk behaviors that are analyzed in the test.

The study’s main objective is to evaluate the psychometric properties—construct validity, predictive validity, internal consistency—and the effectiveness of the RSDED. We proposed to know at the same time the SLV related to risk factors conducive to the development of eating disorders (ED). Further knowledge in this regard will provide information for the prevention of these factors. The effectiveness is estimated looking for: (1)the valid prediction—the presence or not of risk; (2) membership classification—with the least possible error—of the users to these groups (with/without risk of ED), and (3) the frequency of the expected responses—the product of the user–instrument interaction: (1) awareness of body image and eating disorders (realizing); (2) the need to know more about the issue and solve it (problem-solving); (3) self-assessment of own risk habits and behaviors (confirming suspicions); (4) recognition of the need to request a support (seek help); (5) the motivation to change (seek change). The RSDED test format informs users whether they are or are not at risk of developing ED. Its informative capsules—PPT for psycho-educational purposes—contribute to the self-assessment, assisting the user in the SE/SA process. Expected responses are estimated (a) with a group of statements (b) by the frequency of occurrence of those responses in the users’ feedback content through their comments and suggestions.

It should be noted that since the theoretical–practical background of our study was so scarce, the formulation and testing of hypotheses was postponed until the next stage of research.

## 2. Methods

### 2.1. Participants

The non-probability sample was comprised of *n* = 541 women aged 15 to 25 (mean 18.3; SD = 2.4); 61% of the participants resided in Mexico City, 31% in the State of Mexico, and the remaining 8% came from Mexico’s other states.

### 2.2. Instrument

EFRATA. Attitudes towards eating and normal/abnormal eating behaviors were estimated with the EFRATA scale [28], the original scale composed of seven factors with 54 Likert-type items with five response options ranging from “never” = 1 to “always” = 5; the scale explains 43% of the variance with an alpha of 0.89. More information on the EFRATA can be found in 28. The EFRATA has been applied on different occasions [29,30,31,32] in different countries (Mexico, Argentina, Spain, Ecuador).

RSDED. Construct validity was obtained with the exploratory factor analysis EFA and confirmatory factor analysis CFA. The CFA confirmed the results yielded by EFA (Appendix A). This instrument consists of twenty sociodemographic-lifestyle variables (SLV) and eight different modules. Each module is followed by a feedback loop where the score obtained informs whether the results indicate risk/no risk. In the case of risk, information related to the problem is provided, e.g., clarifying its frequency, possible consequences, and associated comorbid risks. It is suggested that the user seeks help and provides information on where to find it. In the case of no risk, healthy behavior is reinforced through congratulations and highlights the values of the user’s actions. After providing feedback for the eighth and last time, concluding the test, a small group of statements appears as free-response material with the following five statements that seek to confirm the presence of the expected answers. The questions and your answers: (1). Allowed you to discover and become aware that you have problems with your body image and your way of eating (realizing); (2). Made you feel uncomfortable (taking you out of your comfort zone), making you think you should do something to solve your problems (problem-solving); (3). Led you to confirm what you already knew or suspected about your body image and your way of eating (confirming your suspicions); (4). Gave you the need to seek help (seek help); (5). Made you feel motivated to change (seek change).

Finally, two open questions appear, asking if the user would recommend the instrument to another person and for comments/suggestions to improve the questionnaire. A high percentage of response was found which can be considered as good reception and interest in the instrument [33,34]. The content of this information, together with the statement group responses, could be used to determine the presence of expected responses.

### 2.3. Procedure

In general, the same procedure was followed as that of the first phase of the preliminary study [35]. The instrument was uploaded to the UNAM platform for public national access via the Internet. Data collection was conducted during 2019. When opening the website, the informed consent section appears, acceptance of which allows access to the capsules—videos for psycho-educational purposes—that address issues related to body image, social media, eating disorders, and healthy diets. The capsules facilitate the SE/SA process, providing information about psychosocial–cognitive–behavioral aspects involved in gestation risk factors. Psycho-educational character refers to a specialized education with psychosocial objectives oriented toward behavioral changes [36].

### 2.4. Statistical Analysis

Descriptive statistics were used to analyze the distributions of sociodemographic - lifestyle variables (SLV) and t-tests to differentiate the means of the groups at risk/without risk of developing ED (SPSS v. 22) (IBM, Mexico, Mexico). For construct validity, exploratory factor analysis (EFA) was applied using the unweighted least squares method with VARIMAX rotation, and confirmatory factor analysis (CFA) was used with the maximum likelihood method with VARIMAX rotation. Discriminant analysis (DA) in steps (forward) was selected because of (1) its predictive function of outcomes and classification of cases into groups, its theoretical possibility to prove whether the cases were classified as they were predicted, and the usefulness of R^2^ related to the size of the effect and (2) the possibility to use a categorical dependent variable (DV) and an interval independent variable (IV), as well as the facility to overcome the no equivalence among variance/covariance matrices, with a large (N) and a group size of at least five times the number of the IV. The DA was applied to the RSDED test to analyze the factors (IVs) related to risky eating behaviors and those of the SE/SA statement group and determine the level of errorless classification of the users’ membership to the at-risk/without-risk of ED groups. The logistic regression analysis (LR; 95% CI) was applied in steps to the SLV data (and those of the SE/SA scale) to determine the relationship between them and the predictive risk factors. A quantitative/descriptive analysis was carried out with the statement group data along with a content frequency analysis with the users’ comments/suggestions.

## 3. Results

### 3.1. Psychometric Properties and Predictive Power of Risk of Developing Eating Disorders

Construct validity was obtained with the exploratory factor analysis EFA and confirmatory factor analysis CFA. The CFA confirmed the results yielded by EFA: three factors were obtained with an overall alpha of 0.91 and RC = 0.93; total variance explained 38%; with 18 items. Risk eating behavior (REB alpha = 0.88, CC = 0.84, EV = 15.66%); restricted diet (RD alpha = 0.85, CC = 0.81, EV = 12.42%); concern over weight and food (CWF alpha = 0.81, CC = 0.71, EV = 10 %) (Appendix A). The data were subjected to a discriminant analysis (DA) to establish their predictive power of risk. Before applying this statistical test, the data for each factor were classified into two groups, without risk and with risk, using the 25th and 75th percentiles as cutoff points. Differences between the means of these groups were obtained, and all comparisons had statistically significant outcomes.

The Box’s M test results could not confirm the homogeneity of covariance—equality between the matrices of the factors—but according to several authors, this test is hypersensitive to slight heterogeneity and small deviations from normality [37,38]. However, DA is a robust test and has the advantage that a large N downplays such rejection. This criterion (large N) was met in this study. The canonical discriminant function revealed, for each factor, eigenvalues ≥ 2.0. Regarding the standardized coefficients, the partial contribution of each variable to the discriminant function, it was found for the REB factor that the variables that most contributed to differentiate between risk/without risk groups were “when others do not see me, I eat more than when I am alone” (0.31) and “I think I eat more than others” (0.32). The CWF variables were “age of onset of weight concern” (0.59) and “weight concern and what is eaten” (0.43). For RD, the highest standardized coefficient (0.94), was the one that contributed the most to the differentiation between the groups, corresponding to the question “what have you done to lose weight?”

Regarding the canonical correlation (CR), a significant association between groups (risk/no risk) with the predictive factors and values of R ≥ 0.80 was obtained (Table 1). When analyzing the ability to explain the variance of each predictive factor and the unexplained ones, similar results were found. The effect size (R^2^) determining coefficients was 0.69 for REB and 0.67 for both CWF and RD. Given that mathematically lambda is equal to 1 (total explained variance), for REB, it was λ = 0.30, while for the other factors (CWF and RD), it was 0.33 for each.

### 3.2. Predictor Variables of Each Factor and Percentages of Correct Classification

The number of risks, predictive variables ≥ 0.30, grouped into each factor was determined by analyzing the factor loads of the DA structure matrix. In the discriminant function of the REB factor, 14 variables remained (five were eliminated), with α = 0.91; the CWF factor was composed of seven variables—only one was left out—with α = 0.81; and the RD factor had four variables (one was eliminated) and α = 0.59. The values of the centroids (means) of each factor showed, in each case, good inter-centroid separations, indicating that the function separates adequately between risk and no risk of developing ED. Table 1 reports the correct classification percentages: the REB factor obtained the highest percentage in the original classification, 97% vs. 96% in cross-validation; the CWF and RD factors showed percentages of 94% and 96%, respectively, correct classification in the original grouping, while in cross-validation they were 93% and 96%, respectively (Table 1).

### 3.3. Self-Exploration/Self-Awareness (SE/SA) in Adolescent Girls

As noted above, the SE/SA statement group, together with the two open questions, could be freely answered. Of the total sample (*n* = 541), 30% (*n* = 162) responded to the five affirmations of the SE/SA statement group, and 55% (*n* = 301) provided comments/suggestions. DA was applied to the SE/SA statement group. The results showed: (1) a significant discriminant function, with acceptable effect size and low proportion of unexplained variance (canonical R = 0.80, *p* ≤ 0.001, R^2^ = 0.64, λ = 0.36) (Table 2), and (2) the percentage of original grouped cases that were correctly classified was 94%, and the cross-validity yielded a percentage of 93% correct classification of membership in the SE/SA statement group (Table 1). The comparison between the users who responded vs. users who did not respond to the SE/SA statement group showed no significant differences. Four variables formed the final SE/SA statement group; the eliminated variable was “confirming your suspicions” because the DA structure matrix showed a factor load less than the cutoff point usually accepted (0.30). The internal consistency of the SE/SA statement group was α = 0.76.

### 3.4. Sociodemographic-LifeStyle Variables (SLV) and Risk Factors of Eating Disorders (ED)

The following variables were selected from the twenty SLV that make up the RESDED to establish the relationship between SLV and risk eating behavior (REB), restricted diet (RD), and concern over weight and food (CWF): age, sex, place of residence, education level and occupation of parents, filial order, age of menarche, active sexual life, age of first sexual experience, the condition of it (consensual/forced), and chemical (hunger inhibitors) and drug consumption. Significant associations were found between (1) smoking and REB (*p* = 0.05, OR = 2.16, 95% CI = 1.01–4.63); (2) smoking and CWF (*p* = 0.01, OR = 2.11, 95% CI = 1.19–3.76); (3) smoking and RD (*p* = 0.005, OR = 2.57, 95% CI = 1.33–4.96), alcohol consumption (*p* = 0.042, OR= 2.11, 95% CI = 1.02–4.31), and first sexual experience before age 15 (*p* = 0.002, OR = 3.58, 95% CI = 1.59–8.07); (4) birth order and REB (*p* = 0.01, OR = 2.9, 95% CI = 1.25–7.08) (Table 3). All the models involved in the described relationships showed adequate goodness of fit although low percentages of explained variance (8–19%). It should also be noted that the increases shown by the ORs were significant, but small, according to the ranges proposed by some researchers [39].

### 3.5. Sociodemographic-LifeStyle Variables (SLV) and Rates of Comments/Suggestions

We looked for significant differences between users who gave/did not give comments/suggestions and sought to recognize what these variables were. The results of the logistic regression showed that the users who gave comments/suggestions were those whose mothers had a higher level of education (preparatory to graduate) vs. a lower level (primary to secondary) (*p* = 0.021, OR = 2.28, 95% CI = 1.13–4.59); those who had started their sex life before age 15 (*p* = 0.003, OR = 5.41, 95% CI = 1.76–16.68); and those with experience of forced sexual intercourse (*p* = 0.03, OR = 0.73, 95% CI = 0.54–0.97) (Table 3). The goodness of global adjustment was 70.8, with 13% of variance explained. It also can be said that the RSDED test received good reception among the users: 42% showed total acceptance (e.g., “consider this is a very complete, clear and precise instrument”; “is excellent”; “should give it a greater diffusion”); to this, we must add the acceptance of the 19% that were classified in the SE/SA category (e.g., “it is excellent because the questions allow us to know what we want, we need, and we are”; “it is a handy tool to seek help if you had the need and it gives you the confidence to answer with the truth”). The remaining 39% was divided into suggestions for improving the instrument (34%) and non-compliance responses (5%), e.g., “it is simple, it says nothing new”; “the results do not seem to match what I answered” (See Appendix B).

## 4. Discussion

The study’s central objective was to evaluate the psychometric properties—construct validity, predictive validity, internal consistency—and the effectiveness of the RSDED. At the same time, it was considered important to know the relationship between risk behaviors and some sociodemographic - lifestyle variables in order to obtain more information to contribute to the prevention of ED in the young student population. It was stated as a theoretical/practical basis that self-detection would occur through a process of self-exploration/self-awareness (SE/SA), a product of the free anonymous interaction between the user and instrument, of the disinhibition of the response product of the online procedure, and the composition of the instrument with feedback spaces and capsules for psycho-educational purposes. The capacity of predictive validity—the presence or not of the risk of developing ED—and the power of correct classification—membership of users in the at-risk/without risk groups—formed the quantitative analysis of the two proposed dimensions to evaluate the effectiveness of the RSDED test. The other, a quantitative/descriptive analysis, was tested through the answers given to the SE/SA statement group and from the content of the open questions, the presence of the expected answers related to realizing, seeking help, motivating change, solving problems, and confirming suspicions of risk behaviors.

### 4.1. The Psychometric Properties and the Predictive Power of Risk and SE/SA Statements

The instrument was shown to have three factors risky eating behavior (REB), concern about weight and food (CWF), and restricted diet (RD) with high values of construct validity, internal consistency, and reliability coefficients, i.e., with acceptable psychometric properties. Regarding the predictive power (contributed by the DA) of the RSDED test, it is essential to note that a fair proportion (a little more than two-thirds) of explained variance or effect size was obtained. It can be said that predictive validity was met and that the RSDED factors show an acceptable effect size (R^2^). Likewise, the high power of correct classification of the factors should be highlighted. We can confirm that the factors of the RSDED test reach a high classification capacity with minimum error. Concerning the statement group, responses were a free choice, answered by those who wished to do so, since they were presented at the end of the RESDED. With this statement group initially composed of five statements and with the material provided by the users’ comments and suggestions, the frequency of the expected responses was obtained from the descriptive aspect (Appendix B). These are indicative of the effect sought by the RSDED, leading to self-detection of risk/non-risk. The quantitative aspect of the SE/SA is related to its predictive and classificatory capacity of users in risk groups. The SE/SA statement group yielded high and significant values.

The preliminary nature of this study justifies to some extent the simplicity of the analysis used—frequency analysis—for the content, the raw material, provided by the users. However, this analysis made clear the potential for information and knowledge that future research with powerful qualitative statistical methods and analysis will make available to aid the physical and mental health of the young student population. Most importantly, this can be achieved with a low cost and grand scope. It is also essential to further improve the content and data collection method regarding the restricted diet factor to achieve an acceptable alpha value. It is suggested to refine the risk measurement by establishing different classification levels and investigating the potential psycho-inducer or specific facilitator of the SE/SA process and that of the capsules in the self-detection process. In the same way, the following research process should take into account the difficulties pointed out by the users in relation to the use of the continuum of response options—from strongly agree to strongly disagree—on the statement group (SE/SA).

### 4.2. Sociodemographic—LifeStyle Profile and Risk Factors of Eating Disorders

The sociodemographic—lifestyle profile of the users was characterized as follows:

#### 4.2.1. Age and Family Social Structure

The sample was composed of young female students, women with an average age of 18 years, with a preparatory education level (three years after high school). Most of them lived with their parents (nuclear family) or with one of them. The participants were mainly firstborn or younger daughters; the middle daughter’s risk of presenting eating disorders increases almost threefold. The problem faced by the middle child is that growing up among older and younger siblings, he/she has to compete against the advantages of the firstborn and the considerations given to the youngest child [40]. It was found that more than half of the fathers and mothers worked, the former as office employees and the latter as housewives. The educational level of the parents was mainly high school, although a quarter had a bachelor’s degree. It was found that a high level of education of the mother (high school or bachelor’s degree) doubled the likelihood that the participant would provide comments and suggestions compared to those whose mothers had lower levels of education. How can this relationship of educated mothers/participatory daughters be explained? The simplest explanation is when parents’ schooling is high, this contributes to better communication and greater freedom for children with less use of retaliation and physical punishment [41]. Research is needed to investigate the co-intervention of other variables to understand the importance of this relationship better.

#### 4.2.2. Sex Life Development

The sexual development and activity of the participants conform to Mexican biocultural patterns. The age of menarche is between 9 and 14 years, as reported by the National Survey of Demographic Dynamics (ENADID; Spanish acronym) [42]. Likewise, more than half of the girls reported having an active sex life, which coincides with the ENADID report [43], which indicates that 62.3% of young women aged 15–29 have started their sexual life and that approximately one in every three adolescents (29.2%) aged 15–19 have already done so.

It was also discovered that some of the participants were also found to have had their first sexual experience before age fifteen. A small percentage responded that it was at age nine, and that same percentage indicated when asked about the condition (consensual/forced) that it was forced and more than half of the girls that it was consensual. When their sex life was initiated before the age of fifteen, and the condition was not consensual, the expression of comments/suggestions is increased up to five times more in the case of the first variable and less than one point for the latter. These variables add to the mother’s high educational level to explain the participatory activity of young women. As can be seen, when the mother’s schooling is high, this contributes, among other things, to the daughters’ ability to express their ideas and ways of thinking. This capacity is a product of the education received in which parenting styles or socialization methods reward assertive behavior, active expression of feelings, desires, and needs, provide support, reinforce reasoning, and use less physical punishment [41,44].

It is also necessary to add the influence of the presence of inadequate sexual experiences, due to their occurrence at an early stage of life [45,46,47], plus the ED with which it is associated because it should not be forgotten that early sexual onset also contributes—along with the consumption of tobacco and alcohol—to the development of a restricted diet [48,49,50]. All this together can represent a source of latent conflict in the face of the stimuli caused by the context of the RSDED test, and because of its conditions of anonymous response, it can be expected that the conflicting content will manifest itself. For example, among the girls’ suggestions were to add questions about personality and depression, socioeconomic situation, feelings of depression, what society says about women’s bodies, and so on (See Appendix B). There is a need to carry out more research to identify the intervention of other variables and the interactive and mediating role they play together.

#### 4.2.3. Drug Consumption

Almost all of the girls responded negatively regarding drug use, so it can be said that the sample was made up of teenagers with no drug problems. This statement’s veracity is high if it is analyzed in light of the “disinhibition effect”, since this effect is due to the condition of anonymity attributed to distance instruments [17,18]. When asked about smoking and alcohol consumption, a little more than a quarter responded affirmatively to the former and more than half of the sample to the latter. These results confirm the interpretation indicated above. We can state that the sample under study was formed of adolescents who do not use drugs other than tobacco or alcohol. When the factors REB, RD, and CWF were analyzed, significant results were found in all three cases with tobacco use; if the participant smoked, the REB and CWF increased twofold and RD by almost three times compared to the results of non-smoking participants. The latter also increased twofold if the participant consumed alcohol in addition to tobacco. These results are consistent with those of previous studies [8,9] in which the relationship between ED and substance-use-related disorders has been given particular importance, e.g., tobacco use for weight control purposes [42].

Finally, from the results of the present study, it can be said that the risk of ED is higher if the girls smoke, consume alcohol, or have early non-consensual sexual experiences. It should be noted that it was expected that the OR increases in risk factors would be relatively small, taking into account the feasible intervention of other variables, which should be defined throughout the process of future research. For the time being, and as a product of this study, it is concluded that distance evaluation yields results with greater validity (confirming what has already been pointed out [2], especially for questions for which there are traditionally false answers or no answers at all, as would be the case with those related to sex and drugs) [5].

## 5. Conclusions

In summary, the results obtained were given in the expected direction, confirming the effect of online disinhibition [17]. As a product of this study, it is concluded that distance evaluation yields results with greater validity (confirming what has already been pointed out), especially for questions for which there are traditionally false answers or no answers at all, as would be the case with those related to sex and drugs. It should be taken into account that this study also confirmed the online effect, which contributes by facilitating the awareness [18] necessary for the assessment or self-detection of risk that requires the promotion of help-seeking among others [3]. This effect was clearly shown in the comments and suggestions from users of the RSDED test (Appendix B). On the other hand, RSDED has acceptable psychometric properties—validity and reliability. It is composed of three factors, risky eating behavior (REB), concern with weight and food (CWF), and restrictive diet (RD) with a predictive capacity of risky behaviors and classification of membership of users to risk/non-risk groups. The group of statements used to estimate self-exploration/self-awareness showed the same acceptable conditions as the RSDED factors, i.e., predictive and risk classification capacity and a high internal consistency coefficient. In this way, the predictive validity was met and the RSDED factors show a good effect size. Likewise, the high power of correct classification of the factors should be highlighted. We confirmed that the factors of the RSDED test reach a high classification capacity. Finally, we can conclude by noting that this study represents an advance in and a contribution to the field of health psychology, reporting the first empirically supported instrument that, within the limitations given by its preliminary nature, shows the potential capacity to meet the needs of its recipients by providing an initial distance self-diagnosis and by opening a viable means of access to health through the path of prevention.

## Figures and Tables

**Table 1 ijerph-18-04103-t001:** Results of correct classification by predictor factor in original grouping and an across-validity grouping.

	Original Grouping	Across-Validity Grouping
Risk eating behavior	97%	96%
Concern over weight and food	94%	93%
Restrictive diet	96%	96%

**Table 2 ijerph-18-04103-t002:** Canonical correlation values, proportions of explained and unexplained variance, and significance (*p*).

	Canonical R	R^2^	Wilks’ Lambda	*p*
Risk eating behavior	0.83	0.69	0.31	0.000
Concern over weight and food	0.82	0.67	0.33	0.000
Restrictive diet	0.82	0.67	0.33	0.000
Self-exploration/Self-awareness	0.80	0.64	0.36	0.000

**Table 3 ijerph-18-04103-t003:** Significant OR values (95% CI). Relationship between risk factors, comments, and sociodemographic variables.

Factor Name	Variables in the Equation	*B*	S.E	Wald	df	Sig.	Exp(B) (OR)	95% CI of Exp(B)	
								Lower	Upper
Concern over weight and food	Smoking	0.75	0.29	6.50	1	0.01	2.11	1.19	3.76
	Constant	02.64	0.93	8.04	1	0.005	0.07		
Risk eating behavior	Smoking	0.77	0.39	3.91	1	0.05	2.16	1.01	4.63
	* Birth order (middle daughter)	1.09	0.44	6.09	1	0.01	2.98	1.25	7.08
	Constant	−5.98	1.42	17.85	1	0.000	0.01		
Restrictive diet	Age of first sexual experience	1.28	0.41	9.47	1	0.002	3.58	1.59	8.07
	Smoking	0.94	0.33	7.98	1	0.005	2.57	1.33	4.96
	Drinking alcohol	0.74	0.37	4.13	1	0.04	2.11	1.03	4.32
	Constant	−4.05	1.10	13.43	1	0.000	0.02		
* Comments	Mother’s education level	0.82	0.36	5.31	1	0.02	2.28	1.13	4.59
	Age of first sexual experience (AFSE)	1.69	0.57	8.65	1	0.003	5.41	1.76	16.68
	AFSE (forced or consensual)	−0.32	0.15	4.65	1	0.03	0.73	0.54	0.97
	Constant	−0.97	1.05	0.87	1	0.35	0.38		

Note: Variables introduced in step 1 for each factor: mother’s occupation, father’s occupation, father’s education level, mother’s education level, age of first sexual experience, first sexual experience was …, smoking, alcohol intake, drug use, and birth order. * Comments refer to material freely contributed by users when asked if they would recommend the instrument and asked for comments and suggestions for improvement.

## Data Availability

Due to research ethics approval, data is not publicly available. The corresponding author can be contacted for further information.

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
