# Peer review of "Online Test of Risk Self-Detection of Eating Disorders"

_ijerph, 2021, doi:10.3390/ijerph18084103_

Round 1

Reviewer 1 Report

Interesting research that provides relevant data for the prevention of Eating Disorders.  Despite this, I have some comments mostly related to a significant lack of scientific references that contextualize and support the results of the research:

The introduction is not very well formulated. The first paragraphs seem more like a series of assumptions by the authors than the results of previous studies. In this sense, it is necessary to provide the bibliographic references that support them.

Besides, an update of the state of the art is recommended in the subsequent paragraphs. The referenced bibliography is often obsolete and leaves aside the most recent studies carried out on the subject.

Section 4.2 should be reformulated in terms of writing. The last paragraph (274-278) is too imprecise to be considered a rigorous statement and contributes to creating a sense of uncertainty around the results.

It is also recommended to deepen the conclusions of the research. They must be supported by the results, specifying the correlation between results and conclusions more clearly and rigorously.

Reviewer 2 Report

Dear authors, the study is very interesting and opens discussion for further studies in the are using online tests and research. I have some comments to improve the manuscript. Methods line 99 - add in Mexico at the end of the line. Line 117 to 118 - is it result? Was the study approved by an Ethics commission? Results lines 150 to 154 - this part belongs to the methods section Line 155 - smoking and REB - p=0.05 What is the p-value considered to be significant? Discussion Parts of the discussion section should come in the results section. An example is from lines 287 to 291 - there isn´t discussion, but data is provided. lines 331 to 335 - it should be in the conclusion section Lines 337 to 347 do not bring discussion related to other studies. it should be in the results section. Discussion is needed for the internal consistent of SE/SA A conclusion section is necessary even if the study brings initial results.

Reviewer 3 Report

It is relevant study aiming to create and test online screening tool for the self-detection of disordered eating and eating disorders in adolescent girls and young women of Mexico.

However, the study has major methodological limitations suggesting that the publication of this manuscript is problematic.

My main concern is the lack of evidence why it is needed to create new online screening instrument instead of using already developed instruments for disordered eating or body image concerns, i.e. EDE-Q, EAT26, BD, DT, MBSRQ or others with the sound psychometric properties? Authors state that “in the construction of the RSDED test the risk factors (REB, RD, CWF) were taken from the EFRATA Scale”. However, there is no example of EFRATA scale and newly developed online self-screening instrument is not provided. It is also unclear why authors analyse associations between different RSDED factors and lifestyle instead of implementing traditional validation procedures of newly developed online screening instrument. There is no evidence in the manuscript that RSDED measures disordered eating or eating disorders since there is no other sound instruments were used in the study and no comparison of data gathered from different measures was provided. Newly developed online instrument should be clearly presented following all procedures of validation. Examples of items or all newly developed instrument should be provided in annexes. The main aim and research question of the study was to assess the effectiveness of the newly developed test. Unfortunately, the main question remained not answered since it is impossible understand from the provided data, how many girls’ and women had disordered eating and how many of them correctly self-detected it by the newly developed measure.

It is also unclear from methods section how the research was organized and how young girls and women have reached the website, how long it was open for respondents, etc.

Introduction part lacks proper referencing and serious edition of English language.

Reviewer 4 Report

In this study, the authors tested the effectiveness and appropriateness of an eating disorders risk self-detection test (i.e. the Risk Self-Detection of Eating Disorders). The test was made available online to adolescent and young adult women. The authors provide strong rationale as to why an online, self-administered test would address challenges related to access to mental health services. In addition, the authors highlight the importance of prevention strategies at early developmental stages. A clear objective is included in the manuscript as well as an explanation for the absence of a hypothesis. The discussion is thorough and confirms previous literature about eating disorder risk factors. Participants found the RSDED generally useful; participants provided feedback and suggestions for improving the RSDED. Overall, the study proposes a helpful, readily accessible tool that could strengthen prevention strategies for eating disorders. A few comments and questions, mainly about the instrument development and study procedure, are provide below for the authors’ consideration.

Introduction

1.  Information about biopsychosocial factors that are associated with eating disorder development is included in the first two paragraphs of the manuscript; however, citations for this information should are missing and should be included.

Method

2. Inclusion/exclusion criteria for taking the test is missing. Also, how was the study advertised?

3. Did participants receive any incentives in exchange for participation?

4. IRB approval information is missing.

5. Line 101: It would be beneficial to describe what the eight modules focus on individually.

6. It is implied, but not clear what type of “risk” the RSDED is detecting – is it detecting risk of developing any eating disorder or specific eating disorders? Or, is it only detecting presence of eating disorder risk factors?

7. It is clear throughout the manuscript that many factors are involved in the risk of developing eating disorders, therefore, an explanation for why RSDED covers specific risk factors should be included. How were the items included in the RSDED selected?

Results

8. The formatting of Table 2 is not aligned and, therefore, difficult to interpret. There is also an extra word (i.e., and) in the second row and a missing word (i.e., awareness) in the last row.

Discussion

9. Lines 300-303 highlight important information related to eating disorders (e.g. childhood trauma and eating disorder development), provide citations where appropriate.

10. In Appendix A, several participants commented that they would have liked to receive links/resources for where to find help. Earlier in the manuscript (i.e. line 105) it is noted that this information was provided to those who were categorized as “at risk” – were any resources made available to those who did not meet “at risk” criteria?

Reviewer 5 Report

The objective of the article is extremely relevant for the intervention in ED, either in traditional treatments or through platforms. In this way, the authors have used a RSDED instrument from which they offer data on its predictive capacity in relation to risk variables in the development of eating disorders.

However, the manuscript should be organized in a different way to improve understanding.

In the introduction add if there are other studies with RSDED. If there are other studies in different countries, this information should be provided in the introduction, as well as the new aspects and the advantages in relation to other questionnaires.

In the participants. It should be more descriptive in relation to the characteristics of the samples and describe the ethical protocol and informed consent, as well as the collection of the sample.

In the description of the instrument:

The construction of RSDED and whether it has been used in other studies or not should also be described. Provide psychometric data to determine its reliability and v alidity, not just its ability to discriminate.

The instruments used to determine risk behaviors and therefore constitute the dependent variable have not been clear to me. They mention EFRATA which should be described in the instruments section.

Round 2

Reviewer 3 Report

The quality of the manuscript was significantly improved at the second round of reviews. There are my remarks for this manuscript.

First, the aim of the study was to evaluate psychometric properties – construct validity, predictive validity and internal consistency and the effectiveness of the new self-detection tool for disordered eating in girls and women of Mexico. However, the main part of the results and discussion sections (line 269-364) is focused on the analysis of the demographic and lifestyle – related criteria (smoking, drinking alcohol, age of the first sexual experience) and disordered eating in women and adolescent girls. These are important results however it is unclear how it is related with the main aim of the study. Sexual development, alcohol and tobacco consumption are related to disordered eating. Therefore, I would recommend to include some evidence on this issue in Introduction section and to revise the main aim of the study.

Line 90. The effectiveness is estimated looking for: 1) the valid prediction – the presence or not of risk; 2) membership classification – with the least possible error – of the users to these groups (with/without risk of ED). What are the differences between these two goals? That need revision or clarification. The presence or not risk – what risk? What does it mean “membership classification” of the users to these groups? It should be clarified here or in Methods section.

Lines 96-100 – this information belongs to Methods section.

Lines 110-116 – EFRATA instrument is presented, however it was not used directly in the present study. Why it is then presented here? I would suggest to clearly present the newly developed instrument informing the readers that the previously validated EFRATA was used as the base for newly developed RSDED.

Line 117 what does it mean FAE and FAC? If it is exploratory and confirmatory factor analysis it should be written in full first mentioned in text and abbreviated as EFA and CFA. I would suggest to include EFA results in appendix A. It would be interesting for international readers to see the content of the subscales of RSDED. Also, it will provide useful information for other researchers aiming to create similar disordered eating self-assessment online instruments.

Line 142. Procedure. I understand that RSDED was uploaded to UNAM for public national access and it is open for public use all the time. The data of 541 women and girls’ is presented in the manuscript. Yet, the question is – what is the time period from which data of the present sample was gathered? If it is from year 2019, it should be clearly stated in the procedure section. Also, I miss the information about the ethical approval for this research.

Presenting results and discussion. As the main aim of the study was to test the validity and effectiveness of the RSDED, I would suggest to provide that information first in the results section and to discuss it first in the discussion section either. EFA, CFA and other analyses of the RSDED might be included in results section, instead of Methods.

It is unclear for the reader what does it mean “Comments” in Table 1. It is not good to present the results in Table 1, then jump to the Table 2 and the come back to Table 1. I would suggest to present information in lines 208-255 before the chapter 3.2.

Line 192. SE/SA scale was not presented as the separate instrument in Methods section therefore the name “scale” makes confusion for the reader. I would suggest not use the word “scale”. Revise the name of the chapter 3.2. Simply write Self-exploration and self-awareness in adolescent girls and women. Or revise the methods (instrument section) clearly presenting the scale as the separate instrument.

Lines 227-229 – this important information belongs to Methods (instrument) section.

Discussion section. No new study results should be provided in discussion section. Put all important demographic information in the section Participants (lines 286-289, menarche age, line 308, sexual life – line 310, 314-315, numbers of drug, alcohol consumption, smoking) and discuss the results without repeating them in discussion section.

Line 386. Conclusions should be provided after the Limitations and future directions.

Reviewer 5 Report

In general, the answers provided in the coverletter should explain the changes made, as well as indicate them.

Introduction: you must increase the number of citations from the last 5 years, eliminating older ones. They must specify the improvements that this questionnaire presents with respect to others.

Instrument: must be exemplified with items from the different modules of the instrument. Also, you need to provide references, where can the instrument be found? It is necessary to be able to replicate the study. If the instrument is published, it must be referenced. If it is not available, it must be indicated and added in an annex.

It must indicate the meaning of the analyzes indicated with the acronyms FAE and FAC (exploratory and confirmatory analysis, it is deduced).

Results:

They should offer the results of VARIMAX rotation, and confirmatory factor analysis.

Discussion: You should comment and discuss all the results, including the analysis confirmatory
